# A Prospective, Multicenter, Randomized, Noninferiority Trial of Stopad^®^ Versus Tachosil^®^ for Hemostasis After Liver Resection

**DOI:** 10.3390/cancers17050757

**Published:** 2025-02-23

**Authors:** Seung Yeon Lim, Gi Hong Choi, Jin Hong Lim, Ho-Seong Han, Yoo-Seok Yoon, Hae Won Lee, Boram Lee, Yeshong Park, MeeYoung Kang, Jinju Kim, Hyelim Joo, Jai Young Cho

**Affiliations:** 1Department of Surgery, Seoul National University Bundang Hospital, Seoul National University College of Medicine, Seoul 13620, Republic of Korea; 82891@snubh.org (S.Y.L.); hanhs@snubh.org (H.-S.H.); yoonys@snubh.org (Y.-S.Y.); lansh@snubh.org (H.W.L.); boramlee0827@snubh.org (B.L.); parkys@snubh.org (Y.P.); mykang@snubh.org (M.K.); 82750@snubh.org (J.K.); kaylin-j@snubh.org (H.J.); 2Department of Surgery, Severance Hospital, Yonsei University College of Medicine, Seoul 03722, Republic of Korea; choigh@yuhs.ac; 3Department of General Surgery, Gangnam Severance Hospital, Yonsei University College of Medicine, Seoul 06273, Republic of Korea; doctorjin@yuhs.ac

**Keywords:** hemostasis, surgical, hepatectomy, carboxymethyl chitosan, batroxobin

## Abstract

Hepatic resection is the most recommended first-line treatment for primary liver cancers and a number of other diseases, but postoperative bleeding has always been an important and serious complication after liver surgery. Stopad is a hemostatic agent utilizing carboxymethyl chitosan and recombinant thrombin-like hemocoagulase (rTLH), which showed promising results in previous studies but is yet to be tested in hepatic resections. We recruited 104 patients and applied either Tachosil (control group) or Stopad (investigational group) after liver surgery and compared their hemostatic efficacy and safety. In this study, Stopad was noninferior to Tachosil.

## 1. Introduction

Primary liver cancer is the sixth most common cancer and the third leading cause of cancer-related deaths worldwide [1]. Although nonoperative management options have evolved markedly over time, resection is still the most recommended first-line treatment for many patients [2,3]. However, postoperative bleeding is an important complication of liver surgery. Blood loss during or after surgery and the need for transfusion are independent prognostic factors for disease-free survival and overall survival [4]. Thus, developing methods to reduce the risk of postoperative bleeding is one of the most important goals for liver surgeons [5].

If oozing bleeding persists even after applying standard operative techniques (e.g., sutures, ligatures, clipping, vascular stapling, electrocautery, argon beam coagulation, etc.), hemostatic products may be the next option. Locally applied pad-type hemostatic products are widely used to control oozing bleeding on the hepatic resection surface because of their efficacy and easy application.

Stopad (NCbit Inc., Seongnam, Korea) is a foam-type pad containing carboxymethyl chitosan (CMCS) and 1 BU/cm^2^ of recombinant thrombin-like hemocoagulase (rTLH). CMCS is a glandular polysaccharide extracted from the carapace of crustaceans. It quickly absorbs oozing fluid and causes a concentration of platelets and red blood cells, which, in turn, facilitates coagulation and is naturally biodegradable [6,7]. rTLH is a recombinant batroxobin that cuts fibrinogen into fibrin by cleavage of fibrinogen A and has potent hemostatic efficacy in both in vitro and in vivo assays [8]. Since CMCS and rTLH have different mechanisms of promoting hemostasis, they are expected to have a synergistic effect, but the efficacy and safety of Stopad is yet to be tested in hepatic resection.

The current study sought to evaluate the hemostatic effect and safety of Stopad during hepatic resection. Tachosil (Takeda Pharmaceuticals, Tokyo, Japan), which contains human fibrinogen and human thrombin, is one of the most popular pad-type hemostatic products. It was selected as a control because its type and purpose of use show the closest similarity to those of Stopad, and its hemostatic effect in hepatic resection has been proven in multiple clinical studies, including at least three randomized controlled trials [9,10,11].

## 2. Materials and Methods

### 2.1. Study Design

We conducted a prospective, multicenter, randomized, noninferiority study of patients requiring open hepatic resection at three domestic institutions (Yonsei University Severance Hospital, Seoul National University Bundang Hospital, and Gangnam Severance Hospital). Subjects were randomized (1:1) to either the investigational group (Stopad) or the control group (Tachosil). The surgeons could not be blinded to the device applied to the subjects, so only the trial subjects were blinded, and randomization and allocation concealment were utilized to prevent bias as much as possible.

The randomization ratio between the experimental group and the control group was 1:1. The permuted block randomization method for each clinical study institution was applied, and the randomization code was generated using the SAS Proc Plan 15.2. Block size and seed number were randomly selected by the investigator in charge of randomization. A randomization envelope for each subject was produced and delivered to the investigator and was opened at the time of randomization for each subject.

Once primary hemostasis was achieved, either Stopad or Tachosil was applied to the resection surface according to the group to which the subject was allocated. Gentle pressure was applied to the pad for 2 to 3 min before visual confirmation of complete hemostasis. If oozing bleeding continued, the process was repeated with a new product for the Stopad group, while no new product was used for the Tachosil group.

The primary outcome was the 3-min bleeding control success rate. The secondary outcomes were the 5- and 10-min bleeding control success rates and the re-bleeding rate. For safety evaluation, all adverse events and possible adverse device effects, including allergic reactions, were categorized using MedDRA, system organ class, and preferred term.

The 3-min bleeding control success rate of Tachosil was 80.7% in a pivotal clinical study conducted by Genyk et al. (2016) for its approval by the United States Food and Drug Administration and 70.6% in a trial conducted by Öllinger et al. (2012) [9,12]. Those studies included 114 and 17 subjects, respectively. The estimated 3-min bleeding control rate for Tachosil, considering the weighted value of the subjects from preceding studies, was 79.4%. Therefore, we set this value as the estimated 3-min bleeding control success rate for the investigational and control devices.

To set the clinically acceptable minimum efficacy rate, we searched for previous studies that validated another widely used hemostatic product, Surgicel (Ethicon, Raritan, NJ, USA). Although its hemostatic efficacy was reported to be a bit lower than that of Tachosil, it is one of the most popular hemostatic products used worldwide. Hence, we postulated that if the investigational device, Stopad, shows comparable efficacy to Surgicel, it can be said that Stopad showed clinically acceptable efficacy. The 3-min bleeding control success rate of Surgicel was 50.0% in a trial conducted by Genyk et al. (2016) and 62.8% in a trial conducted by Bjelović et al. (2018) [9,13]. Those studies included 110 and 113 subjects, respectively. The 3-min bleeding control rate of Surgicel, considering the weighted value of the subjects in preceding studies, was 56.5%. We set this value as the clinically acceptable minimum efficacy rate. Considering these values, we used the difference between the estimated 3-min bleeding control success rates of Surgicel and Tachosil, 22.9% (=79.4% − 56.5%), as the non-inferiority margin in this study.

The number of subjects in each group required to meet the one-sided 97.5% confidence interval and power of 80% was approximately 49. Assuming a 5% early dropout rate, we planned to recruit 52 subjects to each group, making a total subject number of 104. The institutional review board of Seoul National University Bundang Hospital approved the study (IRB No. E-1902-523-003).

### 2.2. Eligibility Criteria

Patients aged ≥19 years who required hepatic resection due to hepatocellular carcinoma, intrahepatic cholangiocarcinoma, hepatolithiasis, liver metastases, or benign hepatic tumors between October 2019 and June 2020 were invited into the trial and underwent screening. Only patients with continuous oozing after primary control of major arterial or venous bleeding with standard surgical techniques during open hepatic resection were ultimately enrolled into the study if they satisfied the other eligibility criteria, described below, and randomized into the Stopad or Tachosil groups.

Patients who voluntarily signed the informed consent form underwent screening. Patients were excluded for the following reasons: emergency operation; coagulation disorder; history of allergic reactions to ingredients contained in Stopad (investigational device) or Tachosil (control device), or to similar substances such as chitosan, crustaceans, batroxobin, anti-human lymphocyte immunoglobulin from horse blood, anti-thymocyte globulin, and freeze-dried Agkistrodon antitoxin; history of alcohol or drug abuse; females who were pregnant, breastfeeding, or who refused to use appropriate contraceptive methods; patients who were deemed inappropriate for this study due to serious complications that occurred during operation; patients who were receiving immunosuppressive treatment; history of liver transplantation; and patients whose surgical risk was considered too high for such studies according to the surgeon’s discretion.

Subjects whose oozing continued after primary control of bleeding without serious surgical complications during hepatic resection were ultimately registered in this trial and randomized 1:1 to either the Stopad group (53 subjects) or the Tachosil group (51 subjects). The participation status of the subjects is summarized in Figure 1.

The subjects were hospitalized for about 7 days after surgery. The study was terminated if no adverse events occurred or if all adverse events had resolved upon completing all monitoring examinations at 30 days after surgery.

### 2.3. Surgical Technique

The routine surgical procedure in all three institutions remained the same during the study period. Liver parenchymal dissection was performed using monopolar and bipolar devices along with a cavitron ultrasonic surgical aspirator (CUSA^®^; Integra, Charlotte, NC, USA). Significant arterial or venous bleeding was primarily controlled with conventional surgical hemostatic techniques such as sutures, ligation, or monopolar/bipolar hemostasis. In patients with continued oozing bleeding even after successful primary hemostasis (e.g., suturing, ligation, vascular clipping, electrocautery, argon beam coagulation, etc.), the random allocation was performed, and either Stopad or Tachosil was applied to the resection surface according to the allocated group, as described in Section 2.1.

### 2.4. Statistical Analyses

For descriptive analyses, the number of subjects, mean, standard deviation, median, minimum value, and maximum value were determined for continuous variables, and the frequency, percentage, and frequency of occurrence were determined for categorical variables.

For continuous variables, the statistical significance of the between-group difference was verified using the independent two-sample *t* test or the Wilcoxon rank sum test, and the statistical significance of the change within each group was verified using the paired *t* test or Wilcoxon’s signed rank test. For categorical variables, the statistical significance of the between-group difference was verified using the χ^2^ test or Fisher’s exact test. The lower bound of the one-sided 97.5% confidence interval was determined for the percentage difference of the primary outcome between the two groups. *p*-values of ≤0.05 were considered significant for all other outcome variables.

## 3. Results

### 3.1. Patient Demographics and Baseline Characteristics

Patient demographics and baseline characteristics are summarized in Table 1. There were no notable differences between the two groups in terms of demographic characteristics, surgical indications, or general medical disorders, as well as history of antithrombotic drugs. However, the Stopad group had more patients with a history of abdominal surgery or interventional treatment (22.64%; 12/53 subjects) than the Tachosil group (9.80%; 5/51 subjects), although the difference was not statistically significant (*p* = 0.0768).

### 3.2. Hepatic Resection Type and Liver Status

Regarding the characteristics of each hepatic resection, the gross morphology of liver parenchyma, hepatic resection site, hepatic resection method, blood flow restriction method, and primary method of bleeding control were evaluated and are summarized in Table 2. There were no statistically significant differences between the Stopad and Tachosil groups in any of the categories.

### 3.3. Efficacy Evaluation

The primary outcome was the 3-min bleeding control success rate. The secondary outcomes were the 5- and 10-min bleeding control success rates and the re-bleeding rate. All 104 subjects who were finally registered and randomized received investigational or control device without any randomization errors.

The primary and secondary outcomes are presented in Table 3. The 3-min bleeding control success rate was 92.45% (49/53 subjects) in the Stopad group vs. 90.20% (46/51 subjects) in the Tachosil group. The lower bound of the one-sided 97.5% confidence level for the difference in success rate between the two groups was −9.82% and was within the preset limit for noninferiority of −22.9%. Therefore, this study proved that the hemostatic efficacy of Stopad was noninferior to that of Tachosil.

Regarding the secondary outcomes, the 5- and 10-min bleeding control success rates were both 100.00% (53/53 subjects) in the Stopad group and were both 98.04% (50/51 subjects) in the Tachosil group. These rates were not significantly different between the two groups (both *p* = 0.4904). Bleeding was not controlled within 10 min in one subject assigned to the Tachosil group; this subject was subsequently treated with an alternative product. The re-bleeding rate was analyzed excluding this subject, and it was confirmed that re-bleeding did not occur in any patient in either group.

### 3.4. Safety Evaluation

The adverse events (AEs) that were reported in 44.23% of subjects (46/104) are presented in Table 4, graded according to the Clavien–Dindo classification system. There were no AEs that led to early dropout through the study. The AEs were reported in 29 subjects (54.72%) in the Stopad group and in 17 subjects (33.33%) in the Tachosil group, indicating a tendency towards a higher incidence of AEs in the Stopad group (*p* = 0.0282). However, although their relationships to the Stopad or Tachosil could not be completely excluded, all AEs were evaluated as being “definitely not related” or “possibly not related” to Stopad or Tachosil, regardless of severity.

There were serious AEs (events that necessitated a longer hospital stay) in 10 subjects in the Stopad group and in 4 subjects in the Tachosil group, which are detailed in Appendix A. The AEs in C-D grade I comprised pyrexia (10/104), delirium (4/104), tachycardia (4/104), and others. The AEs in C-D grade II comprised ileus or nausea/vomiting that required total parenteral nutrition (4/104), portal vein thrombosis (1/104), and ileostomy site bleeding (1/104). AEs in C-D grade IIIa were hematoma (1/104), perihepatic fluid collection (1/104), portal vein thrombosis (1/104), biliary fistula (1/104), adrenal hemorrhage (1/104), and upper gastrointestinal bleeding (1/104). The AE in C-D grade IIIb was wound evisceration (1/104), and the AE in C-D grade IVa was respiratory failure (1/104). None of the adverse events resulted in death, and all of the subjects made a full recovery from the events. No allergic reactions to Stopad or Tachosil were reported.

## 4. Discussion

In our prospective, multicenter, blinded, randomized, noninferiority study including 104 subjects undergoing open hepatic resection, the primary outcome, 3-min bleeding control success rate, demonstrated the noninferiority of Stopad compared to Tachosil (92.45% vs. 90.20%; lower bound of the one-sided 97.5% confidence level: −9.82%). Furthermore, there were no differences in the 5-min bleeding control rate (100% vs. 99.04%; *p* = 0.4904), 10-min bleeding control success rate (100% vs. 99.04%; *p* = 0.4904), or re-bleeding rate (0% vs. 0%). These results indicate that the hemostatic efficacy of Stopad is noninferior to that of Tachosil in open hepatic resection.

Batroxobin, originally isolated from the venom of *Bothrops atrox moojeni*, is one of the most extensively studied snake venoms for its pro-coagulant property. Recombinant batroxobin produced by expression of its cDNA in yeast (*Pichia pastoris*) has been shown to cleave fibrinogen into fibrin and has been demonstrated to reduce bleeding time and blood loss when injected intravenously [8,14]. Moreover, the addition of rTLH to wound dressings has been reported to facilitate hemostasis in a dose-dependent manner [15,16]. On the other hand, chitosan and its derivatives, such as CMCS, are among the most actively studied biocompatible materials that are widely used in various fields, including drug delivery systems, hydrogel scaffolds, cosmetics, and wound dressings [6]. CMCS is a potent absorbent that can rapidly absorb water from blood, creating a localized environment where red blood cells, platelets, and coagulation factors are concentrated, thereby accelerating the coagulation cascade [7]. Additionally, chitosan and its derivatives carry positive charges that neutralize the negatively charged cell membranes of platelets and red blood cells, promoting their aggregation [17,18]. Since rTLH and CMCS possess hemostatic properties through mutually exclusive mechanisms, they were expected to exert a synergistic effect. Indeed, one study has demonstrated that their combined application in wound dressings resulted in improved hemostatic efficacy [16]. Therefore, compared to Tachosil, which contains recombinant human thrombin and collagen, Stopad has theoretical advantages, such as that rTLH has a greater ability to promote fibrin accretion than thrombin, remains effective even in heparinized blood unlike thrombin [19], and that rTLH and CMCS can act synergistically.

The results of this trial demonstrate that the efficacy of Stopad is comparable to Tachosil, which has been reported to exhibit superior efficacy over many other modalities [9,10,11,12]. Furthermore, since numerous chitosan derivatives with various distinct characteristics are already available and can be further optimized through chemical modifications, it may be possible to develop even more effective hemostatic products by exploring alternative combinations of chitosan derivatives. While both rTLH and chitosan have long been recognized for their hemostatic potential, combining the two materials with the concept of synergistically acting hemostatic agents with mutually exclusive mechanisms introduces new possibilities for the future development of powerful hemostatic products. These advancements could contribute to reducing morbidities and mortalities in surgical procedures, particularly in hepatic resections, where bleeding remains one of the most critical complications.

However, it has to be noted that the non-inferiority margin in this study was set somewhat generously. Although Surgicel is a very effective product used worldwide, previous studies have reported that its 3-min bleeding control success rate, reported to be 50.0% in a trial conducted by Genyk et al. (2016) and 62.8% in a trial conducted by Bjelović et al. (2018), is relatively lower than that of Tachosil, reported to be 80.7% in the same trial conducted by Genyk et al. [9,13]. Additionally, patch-type products are thought to have a physical advantage in hemostasis because they have a space restriction effect on the attached surface, giving them an innate advantage in bleeding control compared to gauze-type products like Surgicel. This space-limiting effect may have a greater impact on liver surgeries because the paucity of smooth muscles in hepatic sinusoidal vessels limits its ability to induce vascular constriction, making it much more difficult to control oozing bleeding [20].

Therefore, when setting the non-inferiority margin in this study, it would have been preferable to refer to the hemostatic success rates of a product that, like Tachosil, has a patch form and is as widely used as Tachosil or Surgicel. However, as of now, it is difficult to find a patch-type product that is as widely used as these two, which led to the decision to use Surgicel’s success rate as a reference, resulting in a relatively generous non-inferiority margin. Despite this limitation, the actual results that showed comparable bleeding control success rates of Stopad to Tachosil do support the noninferiority of Stopad.

Another limitation of this study that is noteworthy is that the bleeding control success rate of both Tachosil and Stopad were considerably higher than those reported in previous studies. The 3-min bleeding control success rate of Tachosil, considering the weighted value of the subjects from preceding randomized controlled trials, was 79.4%. However, the 3-min bleeding control success rate in this trial was 90.20% for Tachosil and 92.45% for Stopad. Also, since the 5-min and 10-min bleeding control rates were both 100% for Stopad and both nearly 100% for Tachosil, it was impossible to properly compare their 5-min and 10-min hemostatic efficacy with statistical significance. These high bleeding control success rates might have been caused by the differences in how surgeons define oozing bleeding and the differences in the surgical practices of participating institutions. Such limitations might be overcome by studies with more participants or a different, more rigorous definition of oozing bleeding established by a consensus of the participating surgeons.

In the safety analysis, we found a statistically significant difference in the overall incidence of adverse events. However, all AEs were evaluated as definitely not related or possibly not related to Stopad or Tachosil. Only one serious adverse event was related to operative site bleeding (hematoma) and occurred in the Tachosil group (Appendix A). No allergic reactions to rTLH were observed in any of the subjects in the Stopad group. Therefore, we concluded that the safety of Stopad is not undermined by the relatively higher number of AEs.

The higher number of AEs in the Stopad group could be explained by the higher proportion of subjects with a history of surgical and interventional procedures, including abdominal surgery, portal vein embolization, and transarterial chemoembolization, which can all cause considerable difficulties during surgery because of adhesions and hepatic morphological abnormalities often caused by such procedures. These may lead to longer operation times, more blood loss during surgery, and increased possibilities of other complications. It is also possible that antecedent surgical and interventional procedures had a detrimental impact on the patients’ general condition. However, the difference in the proportion of subjects with a past history of surgical and interventional procedure did not reach statistical significance (22.64% for Stopad group; 9.80% for Tachosil group; *p* = 0.0768), and therefore it cannot be said that the significantly high incidence of adverse events in the Stopad group can be solely attributed to the higher proportion of subjects with past surgical/interventional history. Thus, the safety of Stopad would be more clearly determined if more clarifying data is added by future studies.

## 5. Conclusions

Stopad was noninferior to Tachosil for hemostasis of oozing bleeding that persisted after primary hemostatic control and is safe to use in open hepatic resections.

## Figures and Tables

**Figure 1 cancers-17-00757-f001:**
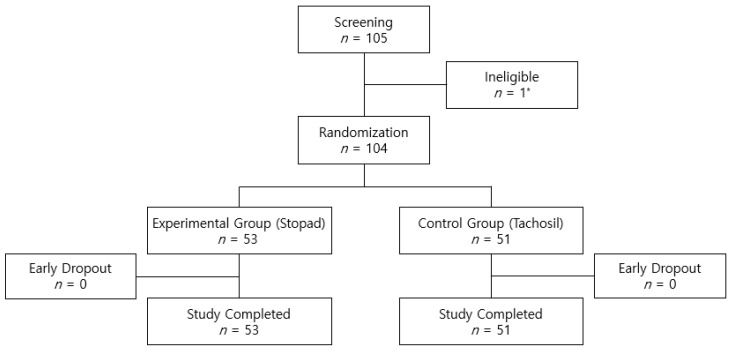
CONSORT diagram of the study participants. * Excluded due to occurrence of serious complication during surgery.

**Table 1 cancers-17-00757-t001:** Patient demographics and baseline characteristics.

		Stopad Group (*n* = 53)	Tachosil Group (*n* = 51)	Total (*n* = 104)
Gender	Male, *n* (%)	36 (67.92)	32 (62.75)	68 (65.38)
	Female, *n* (%)	17 (32.08)	19 (37.25)	36 (34.62)
	Mean ± SD	61.32 ± 11.41	62.25 ± 11.06	61.78 ± 11.20
Smoking history	Yes, *n* (%)	21 (39.62)	21 (41.18)	42 (40.38)
Drinking history	Yes, *n* (%)	15 (28.30)	18 (35.29)	33 (31.73)
Surgical indication	HCC, *n* (%)	20 (37.74)	20 (39.22)	40 (38.46)
	Liver metastases, *n* (%)	15 (28.30)	15 (29.41)	30 (28.85)
	Others, *n* (%)	18 (33.96)	16 (31.37)	34 (32.69)
History of surgical and interventional procedure	Abdominal surgery, *n* (%)	8 (15.09)	5 (9.80)	13 (12.50)
	Interventional treatment of the liver *, *n* (%)	4 (7.55)	0 (0)	4 (3.85)
	Total, *n* (%)	12 (22.64)	5 (9.80)	17 (16.35)
	*p*-value	0.0768		
General medical disorders	Cirrhosis, *n* (%)	5 (9.43)	2 (3.92)	7 (6.73)
	DM, *n* (%)	10 (18.87)	13 (25.49)	23 (22.12)
	HTN, *n* (%)	20 (37.74)	25 (49.02)	45 (43.27)
	DL, *n* (%)	3 (5.66)	2 (3.92)	5 (4.81)
	Coronary artery disease, *n* (%)	2 (3.77)	0 (0)	2 (1.92)
	Chronic pulmonary disease ^†^, *n* (%)	1 (1.89)	2 (3.92)	3 (2.88)
	Chronic kidney disease, *n* (%)	1 (1.89)	1 (1.96)	2 (1.92)
Concomitant drugs	Antithrombotics, *n* (%)	16 (30.19)	18 (35.29)	34 (32.69)

* The interventional procedures comprised three cases of TACE (transarterial chemoembolization) and one case of PVE (portal vein embolization). ^†^ Chronic pulmonary disease comprised two cases of COPD (chronic obstructive pulmonary disease) and one case of pulmonary tuberculosis. Abbreviations: SD, standard deviation; HCC, hepatocellular carcinoma; DM, diabetes mellitus; HTN, hypertension; DL, dyslipidemia.

**Table 2 cancers-17-00757-t002:** Liver condition and surgical procedures.

		Stopad Group (*n* = 53)	Tachosil Group (*n* = 51)	Total (*n* = 104)
Morphology and abnormal type of liver parenchyma	Normal, *n* (%)	33 (62.26)	34 (66.67)	67 (64.42)
	Abnormal, *n* (%)	20 (37.74)	17 (33.33)	37 (35.58)
	Between-group *p*-value		0.6392 *	
Type of abnormal morphology	Steatosis, *n* (%)	5 (25.00)	2 (11.76)	7 (18.92)
	Cirrhosis, *n* (%)	9 (45.00)	7 (41.18)	16 (43.24)
	Fibrosis, *n* (%)	4 (20.00)	6 (35.29)	10 (27.03)
	Other ^‡^, *n* (%)	2 (10.00)	1 (5.88)	3 (8.11)
	Multiple, *n* (%)	0 (0.00)	1 (5.88)	1 (2.70)
	Between-group *p*-value		0.6038 ^†^	
Hepatic resection site	Hemi-hepatectomy, *n* (%)	21 (39.62)	18 (35.29)	39 (37.50)
	Bi-sectionectomy, *n* (%)	4 (7.55)	5 (9.80)	9 (8.65)
	Anterior sectionectomy, *n* (%)	1 (1.89)	2 (3.92)	3 (2.88)
	Posterior sectionectomy, *n* (%)	2 (3.77)	3 (5.88)	5 (4.81)
	Tri-sectionectomy, *n* (%)	6 (11.32)	4 (7.84)	10 (9.62)
	Partial liver resection, *n* (%)	10 (18.87)	8 (15.69)	18 (17.31)
	Segmentectomy, *n* (%)	9 (16.98)	11 (21.57)	20 (19.23)
	Between-group *p*-value		0.9570 ^†^	
Hepatic resection method	CUSA, *n* (%)	30 (56.60)	30 (58.82)	60 (57.69)
	Electrodissection, *n* (%)	19 (35.85)	15 (29.41)	34 (32.69)
	Finger fracture, *n* (%)	1 (1.89)	0 (0.00)	1 (0.96)
	Other, *n* (%)	0 (0.00)	1 (1.96)	1 (0.96)
	Multiple, *n* (%)	3 (5.66)	5 (9.80)	8 (7.69)
	Between-group *p*-value		0.6373 ^†^	
Blood flow restriction method	None, *n* (%)	31 (58.49)	34 (66.67)	65 (62.50)
	Pringle maneuver, *n* (%)	20 (37.74)	17 (33.33)	37 (35.58)
	Selective Pringle maneuver, *n* (%)	2 (3.77)	0 (0.00)	2 (1.92)
	Between-group *p*-value		0.4130 ^†^	
Primary method of bleeding control	Cautery, *n* (%)	12 (22.64)	19 (37.25)	31 (29.81)
	Suture, *n* (%)	2 (3.77)	1 (1.96)	3 (2.88)
	Argon beam coagulation, *n* (%)	10 (18.87)	7 (13.73)	17 (16.35)
	Multiple, *n* (%)	29 (54.72)	24 (47.06)	53 (50.96)
	Between-group *p*-value		0.4341 ^†^	

* Chi-square test. ^†^ Fisher’s exact test. ^‡^ Two subjects in the Stopad group were included in the “Other” category of the abnormal type of liver parenchyma (sinusoidal obstruction syndrome/blue liver and hepatitis d/t chemotherapy). One subject in the control group had cholangitis. Abbreviations: CUSA, cavitron ultrasonic surgical aspirator.

**Table 3 cancers-17-00757-t003:** Bleeding control success rates and re-bleeding rate.

	Stopad Group (*n* = 53)	Tachosil Group (*n* = 51)
3-min bleeding control success rate, *n* (%)	49 (92.45)	46 (90.20)
Difference in bleeding control success rate between the groups (lower bound of the one-sided 97.5% confidence level)	2.26 (−9.82)	
5-min bleeding control success rate, *n* (%)	53 (100.00)	50 (99.04)
*p*-value	0.4904 *	
10-min bleeding control success rate, *n* (%)	53 (100.00)	50 (99.04)
*p*-value	0.4904 *	
Re-bleeding rate, *n* (%)	0 (0.00)	0 (0.00)
*p*-value	NE ^†^	

* Fisher’s exact test. ^†^ NE indicates not estimable. Note: The re-bleeding rate was analyzed after excluding one subject in the Tachosil group who required hemostatic treatment with an alternative product because bleeding continued for more than 10 min.

**Table 4 cancers-17-00757-t004:** Summary of adverse events (AE).

	Stopad Group (*n* = 53)	Tachosil Group (*n* = 51)	Total (*n* = 104)
AE, *n* (%)	29 (54.72)	17 (33.33)	46 (44.23)
*p*-value *	0.0282		
C-D grade I, *n* (%)	19 (35.85)	13 (25.49)	32 (30.77)
C-D grade II, *n* (%)	5 (9.43)	1 (1.96)	6 (5.77)
C-D grade IIIa, *n* (%)	3 (5.66)	3 (5.88)	6 (5.77)
C-D grade IIIb, *n* (%)	1 (1.89)	0 (0.00)	1 (0.96)
C-D grade IVa, *n* (%)	1 (1.89)	0 (0.00)	1 (0.96)
C-D grade IVb, *n* (%)	0 (0.00)	0 (0.00)	0 (0.00)
C-D grade V, *n* (%)	0 (0.00)	0 (0.00)	0 (0.00)

* Chi-square test. Abbreviations: AE, adverse events; C-D grade, Clavien-Dindo grade.

## Data Availability

The data presented in this study are available on request from the corresponding author. The data are not publicly available due to data protection requirements and legal constraints.

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
