# Peer review of "A Prospective, Multicenter, Randomized, Noninferiority Trial of Stopad® Versus Tachosil® for Hemostasis After Liver Resection"

_cancers, 2025, doi:10.3390/cancers17050757_

Round 1
Reviewer 1 Report
Comments and Suggestions for Authors
Seung Yeon Lim et al. present a randomised study comparing Stopad vs Tachosil for hemostasis after liver resection. The rationale to perform this study is given. The methods are adequate. The results need more details. The discussion is too short.
I have some concerns to be addressed:
Abstract
Please specify “with past medical history.”
Avoid expressions like single blind (see below).
The registry says, the trial is still recruiting – please explain or change the registry to recruitment terminated.
Throughout the manuscript: do not call it “experimental and control group”, name them by the intervention.
Methods
Please state clearly how the randomisation sequence was generated and by whom. Furthermore, please state how allocation was performed and when!
Blinding: Avoid the expression single-blinded. Just say blinded and state clearly in the methods section which of the study contributors i.e. patients, surgeons, data collectors, outcome assessors, and data analysts were blinded or not and justify the decision.
In the sample size calculation the product “Surgicel” appears for the first time. This should be explained or changed to Stopad.
I do not agree that 22.9% difference is a clinically acceptable difference. This would mean a number needed to harm of 5 patients, which is too low. You were lucky that the difference was much lower. However, the high assumed difference should be discussed as a limitation.
Results
This is a surgical RCT and therefore, please report postoperative complications according to the Clavien-Dindo classification and calculate a Comprehensive Complication Index (CCI) for both groups and compare it statistically!
Discussion
The discussion lacks a general overview on other trials and does not discuss the limitations of this trial.
Author Response
Thank you very much for taking the time to review this manuscript. Please find the detailed responses below and the corresponding revisions highlighted in the re-submitted files.
Abstract
Comments 1: Please specify “with past medical history.”
Response 1: Thank you for noting the paucity of proper description. We agree with your comment that more specific description is needed. We separated “history of past surgical and interventional procedure” and “general medical disorders” and added detailed list of each category in Table 1. Accordingly, the expression “past history of surgical or interventional procedure” was added to abstract, and the paragraph describing the table was also revised to put more emphasis on this distinction. Please find revised texts in page 1 line 34 (Abstract), page 4 line 28-34 (3.1. Patient demographics and baseline characteristics), and page 5 Table 1.
Comments 2: Avoid expressions like single blind (see below).
Response 2: We appreciate your suggestion, and agree that your expression is much clearer. We deleted every “single blinded” expression in the manuscript, and added more description on the blinding process as well as clearly stating that only the patients were blinded. Please find revised texts in page 2 lines 29-31.
Comments 3: The registry says, the trial is still recruiting – please explain or change the registry to recruitment terminated.
Response 3: We are very grateful that you found this mistake. We immediately requested the registry to change the trial status to recruitment terminated, and the process started but is not completed yet. We promise to follow up to make sure the status is changed as soon as possible.
Comments 4: Throughout the manuscript: do not call it “experimental and control group”, name them by the intervention.
Response 4: Thank you for your suggestion. We changed all the expressions “experimental group” and “control group” to “Stopad group” and “Tachosil group” throughout main manuscript and supplementary material.
Methods
Comments 5: Please state clearly how the randomisation sequence was generated and by whom. Furthermore, please state how allocation was performed and when!
Response 5: We appreciate your point. We added more detailed description of the randomization and allocation process in section 2.1. Study design. Please find revised texts in page 2 lines 32-37.
Comments 6: Avoid the expression single-blinded. Just say blinded and state clearly in the methods section which of the study contributors i.e. patients, surgeons, data collectors, outcome assessors, and data analysts were blinded or not and justify the decision.
Response 6: Thank you very much for kind explanation. As we wrote in response 2, we deleted every “single blinded” expression and added more description on the blinding process as you advised. Please find revised texts in page 2 lines 29-31.
Comments 7: In the sample size calculation the product “Surgicel” appears for the first time. This should be explained or changed to Stopad.
Response 7: Thank you for pointing this out. We agree that this segment needed better explanation. We thought that the minimum clinically acceptable efficacy should be derived from current hemostatic agent that is comparably widely used as the control product Tachosil. Among several candidates, Surgicel was chosen for its consistent popularity over long period of time. This is why the efficacy of Surgicel was mentioned in the manuscript. We added this description in the manuscript. Please find the revised text in page 3 lines 3-8.
Comments 8: I do not agree that 22.9% difference is a clinically acceptable difference. This would mean a number needed to harm of 5 patients, which is too low. You were lucky that the difference was much lower. However, the high assumed difference should be discussed as a limitation.
Response 8: We humbly accept and thank you very much for your insight. We added our thought on the limitations of our study regarding the noninferiority margin, and how it could have compromised the findings of this trial. Please find the revised text in page 8 lines 19-37.
Results
Comments 9: This is a surgical RCT and therefore, please report postoperative complications according to the Clavien-Dindo classification and calculate a Comprehensive Complication Index (CCI) for both groups and compare it statistically!
Response 9: Thank you again for your kind explanation. We revisited the adverse events data and classified the adverse events according to Clavien-Dindo classification as you advised. The results are presented in revised Table 4 (page 7 lines 19-21) and Results section 3.4. (page 7 lines 10-26, page 8 lines 1-6)
Unfortunately, during data collection we counted only the number of patients and number of events for each adverse event item, and we could not find the raw data that specifies which patient experienced which complications. We are sorry to inform you that the lack of this data did not allow us to calculate CCI. Please let us know if you have any other suggestions to strengthen this section.
Discussion
Comments 10: The discussion lacks a general overview on other trials and does not discuss the limitations of this trial.
Response 10: Thank you very much for pointing this out. We added 3 limitation points of this study, including the one you have kindly noted above in comment 8. The limitations were discussed with more detailed values derived from previous trials. Please find added texts in page 8 lines 19-49 and page 9 line 4-17.
Reviewer 2 Report
Comments and Suggestions for Authors
Dear Editor,
Dear Author,
I read and reviewed the present paper submitted to cancers.
A prospective, multicenter, randomized, noninferiority trial of Stopad versus Tachosil for hemostasis after liver resection
It is an interesting article with a recapitulation on treatment options and discussion in therapy of peritoneal carcinomatosis. The austhors provide deep and profound information on pathophysiology and treatment. In my opinion it should be published. It contains important information.
However, I have some comments:
1. In the study design it is necessary to describe properly the intervention group. What patients got and how….
2. Analog the control group!
3. In the chapter statistical analysis the last two parts are confusing. What is this? Power calculation or sample size calculation? Please specify.
4. the first section of the discussion belongs to the methods, or results. Things are here better explain then where they should. Please amend!
5. Please discuss your results. The section discussion contains discussions, in the current manuscript there is summary of design, other methods and results. Please write your manuscript according to common rules.
6. It is sad, results are sounding. The presentation needs amendment.
Author Response
Thank you very much for taking the time to review this manuscript. Please find the detailed responses below and the corresponding revisions highlighted in the re-submitted files.
Comments 1: In the study design it is necessary to describe properly the intervention group. What patients got and how….
Response 1: Thank you for your kind explanation. We agree that the study design needs better description of the intervention. We rearranged the part that describes the intervention each group received from surgical techniques section to study design section. Also, more detailed description of blinding and randomization process was added to this section. Please find the revised text in page 2 lines 29-42.
Comments 2: Analog the control group!
Response 2: Thank you for your observation. We added description of how the randomization and allocation process was done for investigational group and control group in page 2 lines 32-37.
Also, we noted that the Table 1 that compared patient demographics and baseline characteristics of control group and investigational group was not specific enough. Therefore, we revised Table 1 and its description to better emphasize the differences that were found between the 2 groups. Please find the revised Table 1 in page 5 lines 1-7, and revised text in page 4 lines 28-34.
Comments 3: In the chapter statistical analysis the last two parts are confusing. What is this? Power calculation or sample size calculation? Please specify.
Response 3: We are very grateful for your valuable advice. We agree that our manuscript lacked proper explanation of sample size calculation. We added the description of sample size calculation in “study design” section. Please find the revised text in page 3 line 16-18.
Also, we appreciate your insight that the part describing the statistical analysis was misleading. We rearranged the part that describes minimum clinically acceptable efficacy rate and non-inferiority margin to “study design” section, and added explanation on how the noninferiority margin was determined. We hope this helps clarifying this section. Please find the revised text in page 3 lines 3-15.
Comments 4: the first section of the discussion belongs to the methods, or results. Things are here better explain then where they should. Please amend!
Response 4: We appreciate your insight. We rearranged the more detailed description of surgical technique from discussion section to the methods section (2.3. surgical technique). Please find the revised text in page 4 lines 9-12.
Comments 5: Please discuss your results. The section discussion contains discussions, in the current manuscript there is summary of design, other methods and results. Please write your manuscript according to common rules.
Response 5: We humbly accept and thank you very much for your advice. We added our thought on the limitations of our study regarding the noninferiority margin, considerably higher bleeding control success rate compared to previous trials, and significantly higher rate of adverse events in the interventional group (Stopad). Please find the revised text in page 8 lines 19-49 and page 9 lines 4-17.
Comments 6: It is sad, results are sounding. The presentation needs amendment.
Response 6: Thank you for your kind review. We tried to better present the results by changing the classification categories in tables. In Table 1, we specified “past medical history” by separating “surgical and interventional procedure” from “general medical disorder” and add footnotes that lists the type of procedure and illness in each category. Please find the revised Table 1 in page 5 lines 1-7.
Table 4 was changed into a much simpler one that lists number of adverse event in each Clavien-Dindo grade, and the description of specific events in each category was added to the paragraph below. Please find the revised Table 4 and text in page 7 lines 10-26 and page 8 lines 1-6.
Round 2
Reviewer 1 Report
Comments and Suggestions for Authors
All queries were resolved.
Author Response
Comment 1: All queries were resolved.
Response 1: We are so very grateful for your sharp insights and kind suggestions that guided us to greatly improve our paper. Thank you again for the time you have spent for this paper.
Reviewer 2 Report
Comments and Suggestions for Authors
Dear Editor,
Dear Author,
I read and reviewed the present paper (R1-Revisino) submitted to cancers.
A prospective, multicenter, randomized, noninferiority trial of Stopad versus Tachosil for hemostasis after liver resection
My comments:
1. amend the intro of the discussion as follows:
In our prospective, multicenter, blinded, randomized, noninferiority study including 104 subjects undergoing open hepatic resection, the primary outcome, the 3-min bleeding control success rate, reveal noninferiority for Stopad as compared to Tachosil (92.45% vs. 90.20%; lower bound of the one sided 97.5% confidence level: −9.82%)……..
2. please speculate in the conclusion about clinical implications and the potential benefit…
3. Also in the abstract
4. The new preparation of the manuscript is well! Go ahead!
Author Response
Comments 1: amend the intro of the discussion as follows:
In our prospective, multicenter, blinded, randomized, noninferiority study including 104 subjects undergoing open hepatic resection, the primary outcome, the 3-min bleeding control success rate, reveal noninferiority for Stopad as compared to Tachosil (92.45% vs. 90.20%; lower bound of the one sided 97.5% confidence level: −9.82%)……..
Response 1: Thank you for a much more concise, simply better introduction. We amended the first sentence of the discussion as you have suggested. Please find the revised text in page 8 lines 8-11.
Comments 2: please speculate in the conclusion about clinical implications and the potential benefit.
Response 2: We appreciate your suggestion. We elaborated more on the key materials of the investigational device, chitosan (CMCS) and batroxobin (rTLH), to draw more clinical implications and potential benefits of this study by potential future utilization of the materials and exploration of alternative combinations. Please find the revised text in page 8 lines 16-47.
Comments 3: Also in the abstract
Response 3: We added potential benefit of the study to the abstract as you have advised. Please find the revised text in page 1 lines 34-36
Comments 4: The new preparation of the manuscript is well! Go ahead!.
Response 4: We are very grateful for your advice and profound insight, which improved our paper enormously. Thank you also for the encouragement! It is very kind of you! We tried our best to make this paper better, and we hope this revised version meets your expectations. Thank you so much for your time.